# Associations between Self-Reported Anatomical Characteristics of the Penis and Sexual Dysfunction in Men

**Caoyuan Niu [1,2,*]**, **Daniel Ventus [3]**, **Patrick Jern [4]** and **Pekka Santtila [1,2]**

1. School of Psychology and Cognitive Science, East China Normal University, Shanghai 200062, China; pekka.santtila@nyu.edu
2. Faculty of Arts and Sciences, New York University Shanghai, Shanghai 200124, China
3. Experience Lab, Faculty of Education and Welfare Studies, Åbo Akademi University, 65100 Vaasa, Finland; daniel.ventus@abo.fi
4. Department of Psychology, Åbo Akademi University, 20500 Turku, Finland; patrick.jern@abo.fi
* Correspondence: cn2250@nyu.edu; Tel.: +86-18790528652

**Abstract:** Premature ejaculation and erectile dysfunction are common male sexual dysfunctions worldwide, causing substantial distress in men as well as their partners and decreasing the quality and stability of romantic relationships. We investigated the associations between the self-reported anatomical characteristics of penises and sexual dysfunctions in an urban sample of Chinese men. We recruited 1085 Chinese urban men aged from 18 to 50 ($M = 31.37$; $SD = 5.52$) to fill out an online questionnaire regarding the anatomical characteristics of their penis, as well as early ejaculation and erectile problems via two Chinese online survey platforms. The participants reported their age, height, weight, penile length, penile circumference, circumcision status, and foreskin characteristics as well as answered the International Index of Erectile Function-5 and Checklist for Early Ejaculation Symptoms. Both an increasing penile length ($M = 14.49$; $SD = 2.22$) and girth ($M = 15.46$; $SD = 4.36$) were associated with fewer early ejaculation problems, but only an increasing penile length was associated with fewer erectile problems. Less foreskin covering the glans of the penis was associated with fewer early ejaculation and erectile problems. Age was found to have a non-linear relationship with early ejaculation and erectile problems in this cross-sectional study. Specifically, early ejaculation problems decreased until a certain age (31), and then increased with further increases in age. The relationship between the anatomical characteristics of the penis and sexual function is complex. Also, the results suggest that there is a curvilinear non-monotonic relationship between age and sexual dysfunction.

**Keywords:** premature ejaculation; erectile dysfunction; penis size; foreskin; age





## 1. Introduction

Premature ejaculation (PE) and erectile dysfunction (ED), which cause negative outcomes in men and their partners, are common sexual dysfunctions worldwide [1]. A little research has focused on the association between the anatomical characteristics of the penis and sexual dysfunction. In this study, we investigated the associations between the self-reported anatomical characteristics of the penis and PE and ED. Given that age has been previously found to be associated with sexual dysfunction [2,3], as well as there being a reason to believe that age could be associated with how men relate to their sexual anatomy, we also included age in our analyses.

### 1.1. PE and ED

According to the Diagnostic and Statistical Manual of Mental Disorders [4], PE was delineated as a persistent or recurring pattern characterized by the occurrence of ejaculation within approximately one minute following vaginal penetration during partnered sexual activity, occurring prior to the volitional desire of the individual. In this study, PE is

characterized by a lack of control over the timing of ejaculation, a short intravaginal ejaculation latency time (i.e., one minute from the start of vaginal penetration to ejaculation), and subsequent sexual distress [5,6]. On the one hand, men with PE experience lower self-confidence and self-esteem [7,8], more anxiety and depression [9], and interpersonal difficulties [10,11]. On the other hand, PE is associated with less relationship and sexual satisfaction and an increased prevalence of sexual dysfunctions for female partners [2]. On a related vein, one in five women reported that they had broken up or divorced men because of early ejaculation problems in a large sample [12].

ED is defined as the inability to attain or maintain an erection sufficient to obtain satisfaction from sexual intercourse [13]. Also, ED is associated with lower self-esteem and less sexual satisfaction [14]. Several studies have also found that ED is associated with an increased risk of PE [15,16]. A recent study found that one in four men with ED have PE symptoms [17].

## 1.2. Penis Size and Sexual Dysfunction

Penis size is a specific concern related to body image among men as part of men's appearance-related self-esteem [18]. In a sample of 25,594 heterosexual men, 45% suffered from dissatisfaction and anxiety about their penis size, including men with an objectively normal size penis [19].

So far, a little research into the connections between penis size and male sexual function has been conducted. Recently, men have started to seek surgical penile augmentation to increase their penile length or circumference [20,21]. The increases in penile length and circumference after penile augmentation (e.g., the use of allografts, specifically an acellular inert dermal matrix derived from donated human skin tissue, to enhance the circumference of the penis) have been found to improve the participants' sex-related self-esteem and satisfaction with their penis [20]. Further, an increased penile circumference after penile augmentation has been associated with longer ejaculation latency times and better erectile function, probably due to the reduction of penile sensation compared to the baseline data before the surgery [21]. In addition, a recent review indicated that glans penis augmentation reduced PE symptoms [22]. Although they are suggestive of a causal effect of penile length on sexual function, these findings may not be generalizable given that the men seeking surgery may be different from other men. A study of 1027 Egyptian men found that men with ED had shorter fully stretched penis lengths than the men without ED, but no connection between penile circumference and ED was found [23]. However, a study of 689 Brazilian men did not find any association between penile length and erectile function [24].

So far, no studies have directly investigated the link between penis size and sexual function in a non-clinical Asian sample. Based on the available evidence, we expected the men with longer penises to have fewer early ejaculation and erectile problems, while only conducting analyses regarding girth in an exploratory manner.

## 1.3. Penile Circumcision and Sexual Dysfunction

The circumcision of the penis is one of the most common surgical procedures worldwide. The procedure involves the surgical removal of part or all of the foreskin from the penis for, among others, religious, cultural, and medical reasons. Approximately one in three men has been circumcised worldwide [25]. The foreskin is the double layer of skin that covers the glans penis. The possible roles of the foreskin may include keeping the glans moist [26], protecting the developing penis in the womb [27], and enhancing sexual pleasure due to the presence of nerve receptors [28]. A tight foreskin (i.e., phimosis) may cause erectile problems and even pain during sexual intercourse [29], which is a common medical reason for circumcision.

The effect of penile circumcision on sexual function has been investigated for a long time, but remains controversial. The recent reviews indicate that penile circumcision may not have a robust effect on sexual function [30,31], penile sensitivity [30,31], or sexual

pleasure [30,31]. Bronselaer et al. [32] found that male circumcision decreased men's sexual pleasure and orgasm intensity. However, some recent studies have found that circumcised men reported better erectile function and less penile pain at rest and during sex, which might be the reason for the observed improvements of erectile function [29], higher intravaginal ejaculatory latency times, better control over ejaculation, and more satisfaction with sexual intercourse compared to themselves before circumcision [33].

Bossio et al. [34] found that uncircumcised men's foreskin sensitivity to tactile stimulation was higher than that of other penile sites (glans penis, proximal-to-midline shaft of the penis, and midline shaft). Further, penile sensitivity was not different between circumcised and uncircumcised men among the latter penile sites [34]. In conclusion, the foreskin of the penis may be one of the most important sites for tactile stimulation during sex. One possibility is that the reason that circumcision can improve ejaculation control is because of reduced penile sensitivity via removing a part of the foreskin. In conclusion, we assumed that male circumcision improves erectile function (due to unknown reasons) and ejaculation control (due to reduced sensitivity).

We also looked at the differences between men who naturally had different degrees of foreskin covering the glans penis, while the penis had or did not have a full erection. In a flaccid state, a glans penis with less foreskin coverage is more likely to come into contact with undergarments, resulting in friction between the glans penis and the clothes leading to thicker skin, and thereby, the decreased sensitivity of the penis to stimulation, which, in turn, would lead to a higher threshold for ejaculation. Assuming a correlation between foreskin coverage in the flaccid and erect states, we expected the erect state coverage to have the same effect. Also, considering the difficulty with erection if the penis has an excessively tight foreskin [29,34], we expected that the men with less foreskin covering the glans when the penis was erect to have a less-tight foreskin, which would have a smaller impact on the erection, resulting in fewer erectile problems.

### 1.4. Age and Sexual Dysfunction

The association between age and sexual function is still not fully understood. Although some previous research has found that older men have longer self-reported ejaculation latency times [2], age has not always been found to be associated with PE [35,36]. However, the previous studies have relatively consistently found that a higher age is associated with a higher risk of ED [16,37], particularly in men aged over 40 [3]. Also, on the one hand, the higher risk of ED with increased age might also drive an increased risk of PE, as PE and ED are positively associated [15,16,38]. On the other hand, more sexual experience as a function of increased age may lead to less sexual performance anxiety (especially within long-term relationships), which, in turn, might decrease the risk of PE and ED [39].

Interestingly, both cross-sectional and longitudinal studies have shown that testosterone levels decline gradually as men age from their 30s to their 90s [40–42]. Testosterone plays a role in every step of the male sexual response [43,44], with the previous research showing that lower testosterone levels are associated with reduced sexual desire [45] and a higher risk of ED [44,45]. In addition, testosterone replacement treatment can improve the latency times of men with acquired PE (PE appearing only after a man's first sexual experience) [46]. However, a recent review has indicated that the effectiveness of testosterone replacement treatment in improving sexual function is modest and inconsistent, comparable to that of lifestyle interventions [44]. On a similar vein, another contributing factor is cardiovascular disease, contributing to ED during aging [47]. Based on the above, we supposed that the association between age and sexual dysfunction may not simply be linear. Instead, the effects of age on sexual function could be positive among younger men (due to the psychological effects of increased sexual experience) and negative among older men (due to biological changes caused by aging). Therefore, we investigated the association between age and sexual function among adult men, while considering age itself also as a potential moderating factor.

### 1.5. Hypotheses

We hypothesized that men with longer penises would experience fewer early ejaculation and erectile problems and only conducted analyses regarding girth in an exploratory manner (Hypothesis 1).

We hypothesized that circumcised men would experience fewer early ejaculation and erectile problems (Hypothesis 2). We expected that men with less foreskin covering glans of the penis during a full erection would experience fewer early ejaculation and erectile problems (Hypothesis 3).

We expected that the association between age and sexual function is positive among younger men and negative among older men via the increasing sexual experience (Hypothesis 4).

## 2. Materials and Methods

### 2.1. Participants

Adult men who lived in Shanghai, China, were recruited to participate in the survey. Finally, we included 1085 participants who were aged from 18 to 50 ($M = 31.37$; $SD = 5.52$) reported that their biological sex at birth was male, that they were only sexually attracted to women, that their sexual identity was straight, that they had a stable sexual partner, and reported their penile length. We wanted to initially focus on straight men with female partners in order to be able to investigate PE and ED without confounding related to the sexual dysfunction of same-sex-attracted men who were in relationships with women, a relatively common occurrence in China [48]. In this sample, the majority of the participants, 66.5% (722), reported having an undergraduate level of education. A smaller proportion, 22% (239), reported having a postgraduate level of education, while 7.8% (85) reported having a junior college level of education. Additionally, 1.7% (18) of the participants reported having a technical secondary school education, 1.4% (15) reported having a senior high school education, and 0.5% (5) reported having a junior high school education. Only one participant (0.1%) reported only having a primary school education. The majority of the participants, 713 (65.7%), reported being married.

### 2.2. Measures

#### 2.2.1. Demographic Information

The participants were first asked to answer questions regarding their age, height, weight, assigned sex at birth, sexual identity, whether they had a stable sexual partner or not, and education level. The response options for sexual identity were straight, gay, bisexual, asexual, and uncertain. They also answered a question about the sex of the persons they were attracted to, with the response options being men, women, both, neither, and uncertain. The response options for education level were primary school, junior high school, technical secondary school, senior high school, junior college, undergraduate, and postgraduate.

The participants also were asked to answer the questions "Do you currently use SSRI medication (e.g., fluoxetine, paroxetine, sertraline, citalopram, and escitalopram)?" and "Do you have or have had any history of cardiovascular, endocrine disease, etc.?", with response options limited to "Yes" or "No".

#### 2.2.2. Penile Characteristics

Next, the participants were asked to report their penis length and circumference. There was a picture showing the length and circumference of the penis and instructions for the participants regarding how to measure the length and circumference of the penis (see Figure 1; [49]). The questions were "Do you know your penis length (cm) when you have a full erection?" and "Do you know your penis girth or circumference (cm) when you have a full erection?" The options were "Yes, I know (When this option is selected, a fill-in-the-blank number for length or circumference appears)" and "No, I do not know". The instructions to measure penile length after erection were "1. You should keep standing

and the penis must be parallel to the ground; 2. Try to reduce the impact of pre-public symphysis fat where public hair grows by starting the measurement as close to the base of the penis as possible; 3. Measure along the upper side of the penis from the base and extend to the top". A corresponding instruction was given for measuring the circumference of the penis. Based on the result of previous studies [50–52], for penile length, we removed values lower than 4 cm and higher than 20 cm. For penile circumference, we assumed that values lower than 8.5 cm must be diameters and changed them to circumference using geometry, and removed values higher than 30 cm.

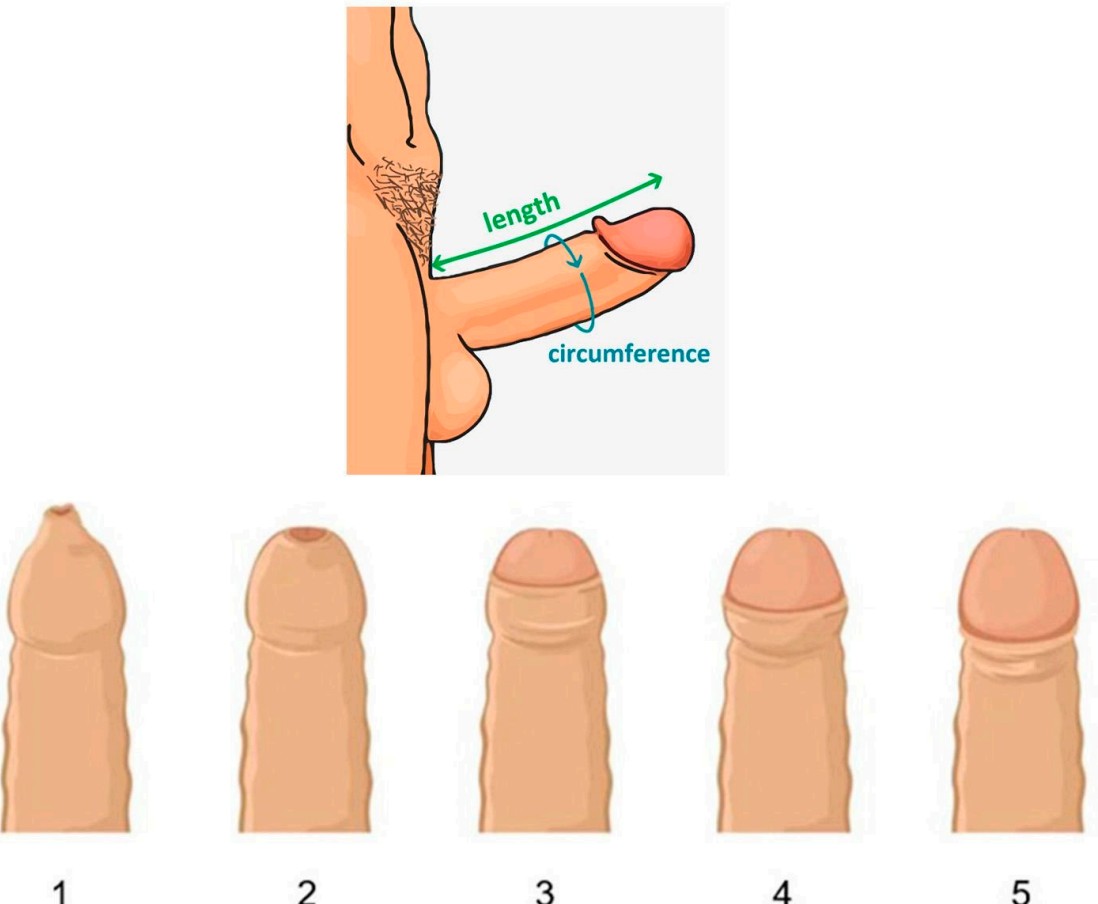

**Figure 1.** The penile size measurement instructions and penile foreskin option image. Note: The upper part shows is the measurement of the length and circumference of the penis. Below is the option indicating the different degrees of the foreskin covering the glans of the penis in a forward direction from the start of the glans of the penis to where the foreskin meets the shaft. Numbers 1–5 indicate the different degrees of the foreskin covering the glans penis. Higher values mean less foreskin is covering the glans of the penis.

Then, participants were asked "Has your penis been circumcised?" If they chose "No", they were asked to answer two questions regarding the degree of the foreskin covering the glans of the penis when they do not have an erection and when they have a full erection. The questions were "To what degree does your foreskin cover your glans penis (the tip of your penis) when you do not have an erection?" and "To what degree does your foreskin cover your glans penis (the tip of your penis) when you have a full erection"? Figure 1 shows the options indicating the different degrees of the foreskin covering the glans penis (100%, 75%, 50%, 25%, and 0%). The participants were asked to choose one of the five options. Higher values mean less foreskin is covering the glans of the penis [53] in a forward direction from the start of the glans of the penis to where the foreskin meets the shaft. In

addition, we re-coded the degree of the foreskin covering the glans penis of participants who reported that their penises were circumcised as 0%.

### 2.2.3. International Index of Erectile Function-5 (IIEF-5)

The International Index of Erectile Function-5 (IIEF-5), which was developed from the 15-item version [54], was used to measure ED [55]. An example item is: "How do you rate your confidence that you could get and keep an erection?" The sum of the five items was computed so that each item was evaluated on a 5-point Likert scale ranging from 1 to 5 for each participant. In a previous Chinese study, the internal consistency assessed with Cronbach's α was 0.790 [56]. The electronic version of IIEF-5 has been found to have excellent internal consistency, excellent test–retest reliability, and convergent validity in Western samples [57]. Cronbach's α was 0.784 for these items in this study. Higher values indicated that better erectile function.

ED was categorized into five distinct classifications according to scores obtained with the IIEF-5 scale. These categories are delineated based on the severity of ED symptoms as follows: severe (scores ranging from 5 to 7), moderate (scores between 8 and 11), from mild to moderate (scores falling within the range of 12 to 16), mild (scores spanning 17 to 21), and no ED (scores ranging from 22 to 25). In this study, we divided the participants into two groups based on the IIEF-5 scale scores: men with ED (scores ranging from 5 to 21) and men without ED (scores ranging from 22 to 25).

### 2.2.4. Checklist for Early Ejaculation Symptoms (CHEES)

The 5-item diagnostic tool Checklist for Early Ejaculation Symptoms (CHEES), which was developed from three earlier ejaculation diagnostic tools and has improved validity [58], was used to measure the ejaculation control of the participants. This measure has been proven to perform well in differentiating men with and without PE in a Western population sample (AUC = 0.98) [58] and to have a good internal reliability of 0.760 [59]. It has not previously been used with Chinese samples. Therefore, we first translated the scale into a Chinese version using translation software. This version was then modified by native Chinese-speaking sex researchers with Ph.D. students. An example item is: "Over the past six months, was your control over ejaculation during sexual intercourse?" The sum of the five items was computed, in which each item was evaluated on a 5-point Likert scale ranging from 1 to 5. In this study, Cronbach's α was 0.776. Higher values indicated more early ejaculation symptoms.

PE was categorized into three distinct classifications according to scores obtained with the CHEES scale. These categories are delineated based on the severity of PE symptoms as follows: strongly indicative of PE (scores ranging from 21 to 25), indicative of PE (scores between 17 and 20), and low probability of PE (scores ranging from 5 to 16). In this study, we divided the participants into two groups based on the CHEES scale scores: men with PE (scores ranging from 17 to 25) and men without PE (scores ranging from 5 to 16).

### 2.3. Procedure

The questionnaire was created on two Chinese survey platforms: WJX and CREDAMO.

On both platforms, the platforms would send an invitation link of the questionnaire to the potential participants who were registered on the platforms, which are adult men who were 18–50 years old and currently living in Shanghai. The potential participants who were interested in participating could click the survey link. After the potential participants clicked the link, they would read the consent form online, and then decide whether they wanted to participate in the study or not and click "Yes, I agree to participate (and confirm that I am eligible for this study)" in case they did. Only the participants who were over 18 years of age, male, Chinese, and had a regular sexual partner were asked to participate. This study was reviewed and approved by the Institutional Review Board. Each participant that responded on WJX was paid from RMB 4 to 7 for their participation, while each participant that responded on CREDAMO was paid RMB 12.5 for their participation, in

line with the internal processes of the two platforms for a questionnaire of this length. The payment procedure was as follows: We paid the data collection service fee to the WJX and CREDAMO platforms. After the participants completed the survey, both platforms would transfer the payment to each participant. The internal processes of the WJX and CREDAMO platforms can prevent multiple submissions from the same IP address that may have been motivated by monetary incentives. Finally, there are 1055 responses from WJX and 996 responses from CREDAMO.

The participants from CREDAMO had a higher age ($t(1974) = 6.394$, $p < 0.001$) than the participants from WJX. We also investigated the stability of the associations between the sexual dysfunction variables and the other variables between the two platforms. The confidence intervals of the correlation between age and the CHEES, the correlation between age and penile circumference, the correlation between CHEES and penile length, the correlation between CHEES and penile circumference, and the correlation between penile length and circumference did not overlap. All the other correlations' confidence intervals overlapped between the two platforms.

## 3. Statistical Analyses

The data analyses were conducted using SPSS 27 and R 4.2.2 [60] and RStudio (2023.6.2.561) [61].

First, we conducted a missing value analysis in SPSS to impute the missing values of penis girth for the participants who had reported the penile length. This was achieved using the quantitative variables of age, height, weight, penis length, and girth using the expectation maximization procedure. In total, 445 missing values were imputed.

Then, we conducted linear regression analyses to investigate whether age, penile length, penile circumference, and the degree of foreskin covering the glans when the penis has a full erection or not were associated with early ejaculation and erectile problems using the interactions package [62].

Next, we conducted moderation model analyses for age as both an independent variable and a moderator in the relationship between age and early ejaculation and erectile problems using the interactions package [62].

Then, we created an age group variable by splitting the data into a younger group (aged 31 or younger than 31; $n = 583$) and an older group (older than 31; $n = 502$) to illustrate the moderation effect. The age of 31 was the median. Finally, we investigated the moderation effect of age groups on the relationship between age and early ejaculation and erectile problems using the interactions package [62].

Finally, we conducted linear regression analyses to investigate whether SSRI medication use and cardiovascular or endocrine disease have an association with early ejaculation and erectile problems [62].

## 4. Results

Table 1 displays the descriptive statistics and the results of the correlation analyses between age, penile length, penile circumference, early ejaculation and erectile problems. A longer penis length and great penis girth were found to be associated with fewer early ejaculation problems, while only a longer penis was associated with fewer erectile problems.

Table 2 displays the result of linear regression analyses between age, penile length, penile circumference, and early ejaculation ($R^2 = 0.102$, $F(5, 1079) = 25.52$, $p < 0.001$) or erectile problems ($R^2 = 0.037$, $F(5, 1079) = 9.276$, $p < 0.001$). A longer penile length (min = 5 cm; max = 20 cm) was found to be associated with fewer early ejaculation and erectile problems. A greater penile circumference (min = 4.7 cm; max = 30 cm) was only associated with fewer early ejaculation problems. Less foreskin covering the glans of the penis was associated with fewer early ejaculation and erectile problems both when the penis has no erection or has a full erection. Separate regression analyses also indicated that circumcision was associated with fewer early ejaculation ($R^2 = 0.037$, $F(5, 1079) = 9.276$,

$p < 0.001$; $B = -1.267$, $SE = 0.171$, $t = -7.405$, $p < 0.001$) and erectile problems ($R^2 = 0.006$, $F(1, 1083) = 7.421$, $p = 0.007$, $B = 0.426$, $SE = 0.157$, $t = 2.724$, $p = 0.007$).

**Table 1.** Descriptive statistics and correlations between age, penile length, penile circumference, and early ejaculation and erectile problems.

| | *n* | *M* | *SD* | **CHEES** | **IIEF-5** | **Age** | **Penile Length** |
|---|---|---|---|---|---|---|---|
| CHEES (Early Ejaculation) | 1085 | 9.63 | 2.85 | | | | |
| IIEF-5 (Erectile Function) | 1085 | 21.71 | 2.55 | −0.544 ** | | | |
| Age | 1085 | 31.37 | 5.52 | −0.094 ** | 0.045 | | |
| Penile Length | 1085 | 14.49 | 2.22 | −0.201 ** | 0.117 ** | 0.008 | |
| Penile Girth | 1085 | 15.46 | 4.36 | −0.194 ** | 0.053 | 0.276 ** | 0.289 ** |

Note. CHEES, Checklist for Early Ejaculation Symptoms. Higher values of CHEES suggest more early ejaculatory problems. IIEF-5, International Index of Erectile Function–5. Higher values of IIEF-5 suggest fewer erectile problems, which also suggests better erectile function. Penile length means the penile length when the penis has a full erection. Penile girth means the penile circumference when the penis has a full erection. ** $p < 0.01$. *p*, probability.

**Table 2.** Linear regression analyses of associations between age, penile length, penile girth, degrees of foreskin covering the glans without an erection and with a full erection, and early ejaculation and erectile problems.

| Dependent | Independent | *B* | *SE* | *t* | *p* |
|---|---|---|---|---|---|
| CHEES | | | | | |
| | Age | −0.023 | −0.016 | −1.474 | 0.141 |
| | Penile Length | −0.191 | −0.038 | −4.935 | <0.001 |
| | Penile Girth | −0.071 | −0.021 | −3.450 | <0.001 |
| | Foreskin Coverage When Penis Not Erect | −0.351 | −0.071 | −4.912 | <0.001 |
| | Foreskin Coverage When Penis Fully Erect | −0.324 | −0.015 | −2.183 | 0.029 |
| IIEF-5 | | | | | |
| | Age | 0.015 | 0.014 | 1.022 | 0.307 |
| | Penile Length | 0.118 | 0.036 | 3.279 | 0.001 |
| | Penile Girth | −0.003 | 0.019 | −0.145 | 0.885 |
| | Foreskin Coverage When Penis Not Erect | 0.140 | 0.066 | 2.109 | 0.035 |
| | Foreskin Coverage When Penis Fully Erect | 0.456 | 0.137 | 3.316 | <0.001 |

Note. CHEES, Checklist for Early Ejaculation Symptoms. Higher values of CHEES suggest more early ejaculation problems. IIEF-5, International Index of Erectile Function-5. Higher values of IIEF-5 suggest fewer erectile problems. Penile length means the penile length when the penis has a full erection. Penile girth means the penile circumference when the penis has a full erection. Higher values of foreskin coverage mean less degree of the foreskin covering the glans of the penis. *B*, coefficient; *SE*, Standard Error; *t*, t-Test; *p*, probability.

Table 3 displays the moderation effect of age and the age groups on the relationship between age and early ejaculation and erectile problems.

The analyses of age as both an independent and moderating variable in the relationship between age and early ejaculation problems ($R^2 = 0.017$, $F(2, 1082) = 10.16$, $p < 0.001$) or erectile problems ($R^2 = 0.004$, $F(2, 1082) = 3.394$, $p = 0.034$) showed that a higher age was overall associated with fewer early ejaculation and erectile problems, but that age also had a significant moderating effect on these relationships.

Analyses using the age group as a moderation variable in the association between age and early ejaculation problems showed that the age group had a significant moderating effect on the relationship ($R^2 = 0.010$, $F(3, 1081) = 6.618$, $p < 0.001$), but not for the relationship between age and erectile problems ($R^2 = 0.027$, $F(3, 1081) = 1.788$, $p = 0.148$). A higher age group was also associated with fewer early ejaculation problems.

Table 4 displays the simple slopes analyses for the relationship between age and early ejaculation and erectile problems in the different age groups. Figure 2 shows the interaction effect between age and the age groups on the relationship between age and early ejaculation problems. In the younger group, increasing age was associated with fewer early ejaculation problems, while the relationship was reversed in the older group.

**Table 3.** Moderation effect of age and age groups on the associations between age and early ejaculation and erectile problems.

| | Group | *B* | *SE* | *t* | *p* |
|---|---|---|---|---|---|
| CHEES | Age | −0.466 | 0.129 | −3.614 | <0.001 |
| | Age: Age | 0.006 | 0.002 | 3.264 | 0.001 |
| CHEES | Age | −0.200 | 0.089 | −2.251 | 0.024 |
| | Age Group | −4.432 | 1.661 | −2.668 | 0.007 |
| | Age: Age Group | 0.124 | 0.053 | 2.323 | 0.020 |
| IIEF-5 | Age | 0.267 | 0.116 | 2.297 | 0.022 |
| | Age: Age | −0.003 | 0.001 | −2.132 | 0.030 |
| IIEF-5 | Age | 0.101 | 0.080 | 1.261 | 0.207 |
| | Age Group | 2.265 | 1.497 | 1.512 | 0.131 |
| | Age: Age Group | −0.068 | 0.048 | −1.331 | 0.184 |

Note. CHEES, Checklist for Early Ejaculation Symptoms. Higher values of CHEES suggest more early ejaculation problems. IIEF-5, International Index of Erectile Function-5. Higher values of IIEF-5 suggest fewer erectile problems. The younger and older groups were split by the median age of 31. Participants aged 31 or younger were defined as younger. Participants older than 31 were defined as older. *B*, coefficient; *SE*, Standard Error; *t*, *t*-Test; *p*, probability.

**Table 4.** Simple slopes results of relationships between age and early ejaculation and erectile problems in different age groups.

| | Group | *B* | *SE* | *t* | *p* |
|---|---|---|---|---|---|
| CHEES | Younger | −0.077 | 0.041 | −1.869 | 0.062 |
| | Older | 0.046 | 0.033 | 1.384 | 0.167 |
| IIEF-5 | Younger | 0.038 | 0.037 | 1.010 | 0.313 |
| | Older | −0.026 | 0.030 | −0.868 | 0.386 |

Note. CHEES, Checklist for Early Ejaculation Symptoms. Higher values of CHEES suggest more early ejaculation problems. IIEF-5, International Index of Erectile Function-5. Higher values of IIEF-5 suggest fewer erectile problems. The younger and older groups were split by the median age of 31. Participants aged 31 or younger were defined as younger. Participants older than 31 were older. *B*, coefficient; *SE*, Standard Error; *t*, *t*-Test; *p*, probability.

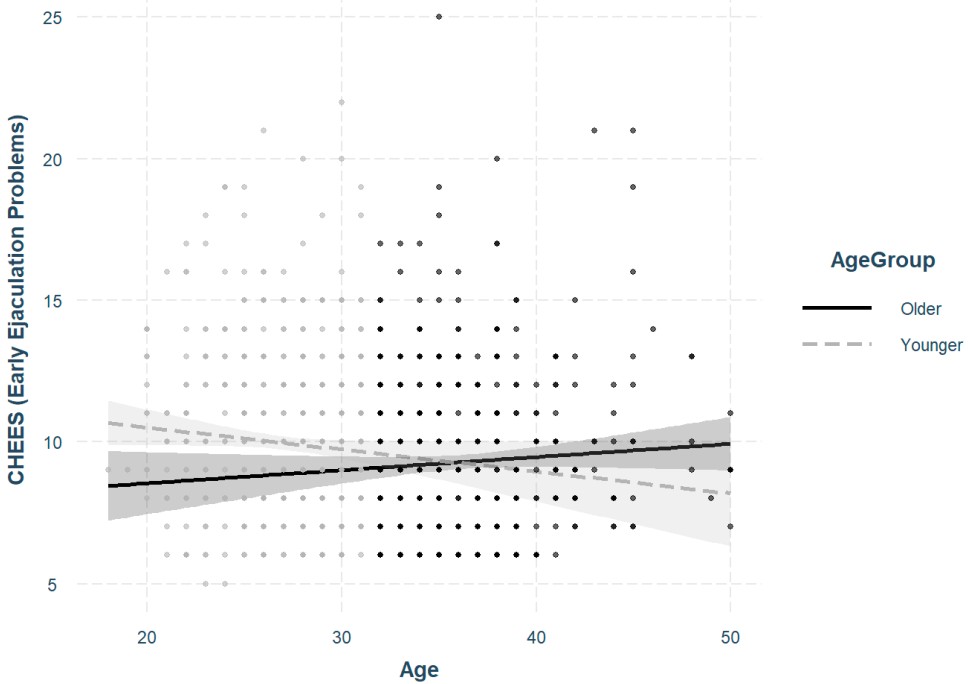

**Figure 2.** The interaction effect between age and age groups on the relationship between age and early ejaculation problems. CHEES, Checklist for Early Ejaculation Symptoms. The different color dots indicated the actual data points of CHEES scale score. The different color shadows indicated a 95% confidence interval.

Table 5 shows the descriptions of the penile sizes in different PE symptom groups. Table 6 shows the descriptions of the penile sizes in different ED symptom groups.

**Table 5.** Descriptive statistics of penile sizes of men who do and not experience premature ejaculation.

| | **Men without Premature Ejaculation** | | | | | | **Men with Premature Ejaculation** | | | | | |
|---|---|---|---|---|---|---|---|---|---|---|---|---|
| | *n* | *M* | *SD* | Median | Min | Max | *n* | *M* | *SD* | Median | Min | Max |
| Penile Length | 1058 | 14.52 | 2.19 | 15 | 5 | 20 | 27 | 13.23 | 2.88 | 14 | 7 | 18 |
| Penile Girth | 1058 | 15.14 | 2.81 | 15.13 | 4.7 | 30 | 27 | 13.85 | 3.34 | 13.81 | 7.9 | 25.1 |

Note. For the men who do not experience premature ejaculation, the CHEES scale scores ranged from 5 to 16. For the men who experience premature ejaculation, the CHEES scale scores ranged from 17 to 25.

**Table 6.** Descriptive statistics of penile sizes of men without and with erectile dysfunction.

| | **Men without Erectile Dysfunction** | | | | | | **Men with Erectile Dysfunction** | | | | | |
|---|---|---|---|---|---|---|---|---|---|---|---|---|
| | *n* | *M* | *SD* | Median | Min | Max | *n* | *M* | *Sd* | Median | Min | Max |
| Penile Length | 692 | 14.65 | 1.98 | 15 | 6 | 20 | 393 | 14.20 | 2.57 | 14 | 5 | 20 |
| Penile Girth | 692 | 15.03 | 2.96 | 15 | 4.7 | 30 | 393 | 15.23 | 2.57 | 15.35 | 7.9 | 25.7 |

Note. For the men without erectile dysfunction, the IIEF-5 scale scores ranged from 22 to 25. For the men with erectile dysfunction, the IIEF-5 scale scores ranged from 5 to 21.

Table 7 shows the result of linear regression analyses between the SSIR medication user status, having or having had cardiovascular or endocrine disease, as well as early ejaculation and erectile problems. The results indicated that currently using SSRI medication and having or having had any history of cardiovascular or endocrine disease were associated with both more early ejaculation and erectile problems.

**Table 7.** Linear regression analyses of associations between SSRI medication use and chronic disease and early ejaculation and erectile problems.

| **Dependent** | **Independent** | *B* | *SE* | *t* | *p* |
|---|---|---|---|---|---|
| CHEES | | | | | |
| | SSRI Medication | 1.996 | 0.531 | 3.762 | <0.001 |
| | Cardiovascular or Endocrine Disease | 1.701 | 0.412 | 4.126 | <0.001 |
| IIEF-5 | | | | | |
| | SSRI Medication | −1.728 | 0.475 | −3.638 | <0.001 |
| | Cardiovascular or Endocrine Disease | −1.568 | 0.369 | −4.25 | <0.001 |

Note. CHEES, Checklist for Early Ejaculation Symptoms. Higher values of CHEES suggest more early ejaculation problems. IIEF-5, International Index of Erectile Function-5. Higher values of IIEF-5 suggest fewer erectile problems. SSRI, selective serotonin reuptake inhibitors. *B*, coefficient; *SE*, Standard Error; *t*, *t*-Test; *p*, probability.

## 5. Discussion

This study investigated the associations between age, penile length and circumference, foreskin characteristics, and early ejaculation and erectile problems. The results indicated that the greater length and girth of the penis were associated with fewer early ejaculation problems, while only a longer penile length was associated with better erectile function. In addition, a complex relationship between age and early ejaculation problems was found.

The average self-reported erect penis was 14.49 cm long. This is slightly shorter than the 16.8 cm reported for a European sample [51], but similar to what has been reported for a US sample (14.2 cm) [50] and other Asian samples (13.0 cm) [52]. The mean erect circumference was 15.46 cm, which is higher than in the above-mentioned Western samples [50,51] and Asian samples [52], which suggested that the mean erect circumference was approximately 11.5 cm. However, the measurement method, social desirability on the part of the reporting man, and the presence of fat tissue can impact the accuracy of the outcomes of penile size measurement [63]. For example, the instructions in this study reduced the participants to keep standing and keep the penis parallel with the ground, while other

studies did not do. Having a longer penis was associated with also having a penis with a greater circumference, which has also been found in previous research [23,63]. However, the correlation between length and circumference was not very strong. A previous study found a longer penile length from the pubis to the distal glans (from bone to tip) was associated with a greater penile circumference ($r = 0.55$, $p < 0.001$), but not for penile length from the suprapubic skin to the distal glans (from skin to tip) [64]. Therefore, in this study, we considered that the weak correlation observed may be attributed to a bias resulting from variations in the measurement of penile length. Specifically, some participants may have measured the penile length from the bone to the tip, while others may have measured it from the skin to the tip.

Men with a shorter penis reported more ejaculatory and erectile problems in this study. It has previously been reported that men with a shorter penis would suffer from more dissatisfaction and anxiety about their penis size [65]. In addition, an insufficient erect penis also may trigger more dissatisfaction with their penis performance. The dissatisfaction and anxiety about penis size can trigger worry and shame regarding body image and sexual performance during sex [66], even body dysmorphic disorder [66]. The balance and timing of the sympathetic and parasympathetic nervous systems are crucial for a well-functioning male sexual response [67]. Excess anxiety about the body and performance during sex can trigger sympathetic overactivity, which might inhibit erections and facilitate ejaculation. This may result in insufficient erections and early ejaculation [24,66,68]. Men with longer erect penises were associated with fewer erectile problems in this study. However, erect penile circumference seems to not have an association with erection, which is consistent with the previous research [23,24].

For younger men (younger than or aged 31 years), an increasing age had an association with better ejaculation control and erectile function. For older men (older than 31 years), increasing age had a negative association with ejaculation control. We assumed that the underlying mechanism may be that in men aged 31 years or younger, psychological factors (i.e., less sexual performance anxiety with increased sexual experience) positively affect sexual function with increased age. However, in men over 31, biological factors (i.e., cardiovascular disease and sex hormone level decline) would start having a negative impact on at least erectile function, overwhelming the benefits from sexual experience. It is less clear why men in the older group would have more early ejaculation problems when getting older. We assumed that the underlying mechanism may be that in men over 31, the physical condition would decrease because the cardiovascular disease risk arises and sex hormone level decline. Sexual intercourse is most physically demanding in terms of lumbar flexion in men [69]. They will experience more physical problems during sexual intercourse (i.e., exhaustion and musculoskeletal pain) because of poor physical condition compared with younger men [70]. Exhaustion and musculoskeletal pain might lead to an early and more sympathetic activity, which could lower their ejaculation threshold and early activate the ejaculatory reflex.

The results indicated that being circumcised might be associated with improved sexual function. Specifically, compared to uncircumcised men, the circumcised men had better ejaculation control and erection function. These results are consistent with a previous study conducted in China [33] and inconsistent with the recent reviews [30,31]. We suggest that the underlying mechanism of the positive effect of male circumference on sexual function may be that the part of the foreskin, which is the most sensitive site of the penis to tactile stimulation during sex [34], was removed. In addition, the exposure of the skin of the glans of the penis to the air and friction with underpants could further reduce the sensitivity of the glans of the penis after circumcision. Therefore, circumcised men have better ejaculation control because of their reduced penile sensitivity. Surprisingly, we also found that circumcised men had fewer erectile problems. The main reason why men in China get circumcised is due to medical reasons (i.e., phimosis) [33]. We assumed that the reason might be the tight foreskin of the penis glans (i.e., phimosis), disturbing the erection, which, in turn, leads to an insufficient erection in some cases. Circumcision involves the

overall or partial removal of the foreskin glans will reduce this effect, and consequently, reduce the erectile problems.

Men with more erectile problems also had more early ejaculation problems, which is in line with the prior studies [15,16,38]. Men with erectile problems have less self-confidence in having a sufficient erection allowing penetration. Then, this lower self-confidence will lead to excessive worry and anxiety regarding the penis size, erections, as well as sexual performance [39,71]. This excessive anxiety will disturb the parasympathetic activity to facilitate an erection and results in an overactive sympathetic nervous system [72–74], triggering a worse imbalance of autonomic nervous activity, which may also be a cause of poorer ejaculatory function in men. In this way, ED symptoms can have a positive association with PE symptoms. However, further studies are needed to understand the details of this process.

The results indicated that men of an older age would have a thicker erect penis, which is in line with a prior study [75]. However, a prior study found a weakly negative correlation between age and flaccid circumference [76]. In addition, some studies indicated that age is not correlated with penile length and circumference [77,78]. It is possible that the reason for this could be fat tissue accumulating in the penis with increasing age. However, there is only a little research to investigate the relationship between age and the fat tissues of the penis.

Interestingly, SSRI medication use was associated with both more PE and ED symptoms. The negative association between SSRI medication use and ejaculation control is not in line with the previous studies which have found SSRI medication (i.e., fluoxetine and paroxetine) to improve the intervaginal ejaculation latency time in men with PE [79,80]. In this sample, most participants using SSRI medication (27 in 29; 93%) were men without PE. Therefore, it may be that the underlying health problem (e.g., depression or anxiety) for which these men were taking an SSRI may be behind the observed negative association. In contrast, it was not surprising to find a relationship between SSRI medication use and poorer erections given that one of the side effects of SSRI is the loss of sexual desire [80,81].

Cardiovascular or endocrine diseases were found to be associated with more PE and ED symptoms. The previous studies have indicated that cardiovascular diseases were risk factors for ED [82], and also for premature ejaculation [83]. A previous study found that compared with males without sexual dysfunction, men with PE have higher levels of testosterone, and men with ED have higher levels of estradiol [84]. The ratio of estradiol to testosterone was also higher in men with PE and ED compared with that of the men without sexual dysfunction [84]. Therefore, the underlying mechanism of the association between endocrine diseases and PE and ED symptoms may be an imbalance between different hormones.

## 6. Conclusions

The relationship between the anatomical characteristics of the penis and sexual function is complex. Also, the results suggest that there is a curvilinear non-monotonic relationship between age and sexual dysfunction.

### *Limitations and Future Directions*

There were some limitations to this study. First, the penis sizes were measured by the participants. The data on penile length was in line with a previous study [52], but the penile circumference was not. The potential limitation of such self-reports is that some participants may not understand the instructions or may, for other reasons, not perform an accurate measurement of their penis size or report the results accurately. Future studies should employ experimenters to measure the penis size who have the related training using the device. Second, we only measured the circumcision statuses of the men, but not the age at which circumcision had been performed. A new review indicated that circumcisions performed in infancy, childhood, and adulthood have different psychological effects on men [85]. Future studies should also measure the age at circumcision. Last but not least,

this study omitted questions that ask men to rate their penis size in comparison with the norm for their population or comparison group, as well as their satisfaction with their penis size. Future studies should have these questions to measure participants' satisfaction and potential anxiety with their penis size to future investigate the underlying mechanism of the association between penis size and sexual dysfunction.

**Author Contributions:** Conceptualization, P.S.; supervision, P.S.; investigation, C.N.; data collection, C.N.; data curation, C.N.; methodology, C.N.; formal analysis, C.N.; writing—original draft, C.N.; writing—review and editing, C.N., D.V., P.J. and P.S.; visualization, C.N., D.V. and P.J. All authors have read and agreed to the published version of the manuscript.

**Funding:** This research was funded by an internal NYU Shanghai grant given to P.S.

**Institutional Review Board Statement:** The study was conducted in accordance with the Declaration of Helsinki, and the protocol was approved by the Institutional Review Board of the New York University Shanghai (protocol code: 2021-054. The survey of the sexual function and satisfaction of Chinese men and date of approval: 12 September 2021).

**Informed Consent Statement:** Informed consent was obtained from subjects, but no signed consent form was used. The potential participants read the consent form online, and then decided whether they would participate in the study or not and clicked "yes, I agree to participate (and confirm that I am eligible for this study)". They did not give any information that could be used to identify them personally.

**Data Availability Statement:** Data available in a publicly accessible repository. The data presented in this study are openly available in [OSFHOME] at [osf.io/t9z7y].

**Conflicts of Interest:** The authors declare no conflict of interest.

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
