# Peer review of "Associations between Self-Reported Anatomical Characteristics of the Penis and Sexual Dysfunction in Men"

_sexes, doi:10.3390/sexes4040040_

Round 1
Reviewer 1 Report
Comments and Suggestions for Authors
The manuscript studies the possible correlation between penile self-reported anatomical aspects and sexual dysfunction in men.
The manuscript is potentially interesting, but the authors need to clarify some aspects.
- While for penile length the data presented are in line with those already present in the literature, the values reported for penile circumference are not in line with those already present in the literature. The subjects studied probably did not perform an accurate measurement. This aspect should be highlighted more clearly in the “Limitations and Future Directions”.
-What elements were decisive for using 31 years as a cut off between younger and older subjects?
-What was your cut-off score for considering a subject to have ED? It is appropriate to draw up a table that divides the subjects studied into "Subjects with ED" and "Subjects without ED" and insert mean values with SD, median and range (minimum-maximum) of penile length and circumference in the two groups.
-What was your CHEES cut-off score to consider a subject as having PE? It is appropriate to draw up a table that divides the subjects studied into "Subjects with PE" and "Subjects without PE" and insert mean values with SD, median and range (minimum-maximum) of penile length and circumference in the two groups.
Comments on the Quality of English LanguageMinor editing of English language are required
Author Response
The manuscript studies the possible correlation between penile self-reported anatomical aspects and sexual dysfunction in men. The manuscript is potentially interesting, but the authors need to clarify some aspects.
-While for penile length the data presented are in line with those already present in the literature, the values reported for penile circumference are not in line with those already present in the literature. The subjects studied probably did not perform an accurate measurement. This aspect should be highlighted more clearly in the “Limitations and Future Directions”.
Response: Thank you for your comment. We have added the above aspect to the “Limitations and Future Directions” (see page 13). The text was “First, the penis size was measured by self-reported. The data on penile length was in line with the previous study (Promodu et al., 2007) but the penile circumference was not. The potential limitation of such self-reports is that some participants may not understand the instructions or may for other reasons not perform an accurate measurement of their penis size or report the results accurately”.
-What elements were decisive for using 31 years as a cut off between younger and older subjects?
Response: Thank you for your comment. There are two elements behind using 31 years as a cut-off between younger and older subjects. First, testosterone plays a role in every step of the male sexual response with previous research showing that lower testosterone levels are associated with reduced sexual desire and more ED. In addition, testosterone replacement treatment can improve the latency times of men with acquired PE (PE appearing only after a man’s first sexual experience). Both cross-sectional and longitudinal studies have shown that testosterone levels decline gradually as men age from their 30s to their 90s. The above text is now in the session of “Age and Sexual Dysfunction” (see page 3). Second, the age of 31 also happened to be the median in this sample. Based on the above, we decided to use 31 years as the cut-off. This element is now mentioned in the session “Statistical Analyses” (see page 7).
-What was your cut-off score for considering a subject to have ED? It is appropriate to draw up a table that divides the subjects studied into "Subjects with ED" and "Subjects without ED" and insert mean values with SD, median and range (minimum-maximum) of penile length and circumference in the two groups.
Response: Thank you for your comment. First, we have added the following description to the end of the session regarding the IIEF-5 scale (see page 6): ED was categorized into five distinct classifications according to scores obtained from the IIEF-5 scale. These categories are delineated based on the severity of ED symptoms as follows: severe (scores ranging from 5 to 7), moderate (scores between 8 and 11), mild to moderate (scores falling within the range of 12 to 16), mild (scores spanning 17 to 21), and no ED (scores ranging from 22 to 25). In this study, we divided the participants into two groups based on the IIEF-5 scale scores: men with ED (scores ranging from 5 to 21) and men without ED (scores ranging from 22 to 25). We have now added a table that divides the subjects studied into "Men with erectile dysfunction" and "Men without erectile dysfunction" and insert mean values with SD, median, and range (minimum-maximum) of penile length and circumference in the two groups (see page 10).
|
|
Men without erectile dysfunction |
Men with erectile dysfunction |
||||||||||
|
|
n |
M |
SD |
Median |
Min |
Max |
n |
M |
Sd |
Median |
Min |
Max |
|
Penile Length |
692 |
14.65 |
1.98 |
15 |
6 |
20 |
393 |
14.20 |
2.57 |
14 |
5 |
20 |
|
Penile Girth |
692 |
15.03 |
2.96 |
15 |
4.7 |
30 |
393 |
15.23 |
2.57 |
15.35 |
7.9 |
25.7 |
Note. Men without erectile dysfunction, IIEF-5 scale scores ranged from 22 to 25. Men with erectile dysfunction, IIEF-5 scale scores ranged from 5 to 21.
-What was your CHEES cut-off score to consider a subject as having PE? It is appropriate to draw up a table that divides the subjects studied into "Subjects with PE'' and "Subjects without PE" and insert mean values with SD, median, and range (minimum-maximum) of penile length and circumference in the two groups.
Response: Thank you for your comment. First, we have added the following description to the end of the session regarding the CHHES scale (see page 6): PE was categorized into three distinct classifications according to scores obtained from the CHEES scale. These categories are delineated based on the severity of PE symptoms as follows: strongly indicative of PE (scores ranging from 21 to 25), indicative of PE (scores between 17 and 20), and low probability of PE (scores ranging from 5 to 16). In this study, we divided the participants into two groups based on the CHEES scale scores: men with PE (scores ranging from 17 to 25) and men without PE (scores ranging from 5 to 16). We have now added a table that divides the subjects studied into "Men with premature ejaculation" and "Men without premature ejaculation" and insert mean values with SD, median, and range (minimum-maximum) of penile length and circumference in the two groups (see page 10).
|
|
Men without premature ejaculation |
Men with premature ejaculation |
||||||||||
|
|
n |
M |
SD |
Median |
Min |
Max |
n |
M |
SD |
Median |
Min |
Max |
|
Penile Length |
1058 |
14.52 |
2.19 |
15 |
5 |
20 |
27 |
13.23 |
2.88 |
14 |
7 |
18 |
|
Penile Girth |
1058 |
15.14 |
2.81 |
15.13 |
4.7 |
30 |
27 |
13.85 |
3.34 |
13.81 |
7.9 |
25.1 |
Note. Men without premature ejaculation, CHEES scale scores ranged from 5 to 16. Men with premature ejaculation, CHEES scale scores ranged from 17 to 25.
Reviewer 2 Report
Comments and Suggestions for Authors
The authors should be congratulated for their work and for addressing an important topic. However, some points warrant mention:
1. once defined an acronym, the full form should be avoided.
2. in the "Introduction" section, the definition of PE is not fully correct. I suggest to refer to official guidelines.
3. In the "Introduction" section, in "1.1. PE and ED", the authors should also present data on the contemporary presence of both ED and PE, such as in PMID: 35915329.
4. among penile girth augmentation techniques, I suggest also to report the benefits of glans penis augmentation, such as in PMID: 34400809.
5. why did the authors choose 31 years to define young vs. old men? Usually, the scientific community uses 40 y.o. as a threshold.
Comments on the Quality of English Language
The English language does not need extensive editing.
Author Response
The authors should be congratulated for their work and for addressing an important topic. However, some points warrant mention:
- once defined an acronym, the full form should be avoided.
Response: Thank you for your comment. We have checked all the acronyms and avoided using the full form again after the abbreviation has been used the first time.
- in the "Introduction" section, the definition of PE is not fully correct. I suggest to refer to official guidelines.
Response: Thank you for your comment. We have added the following official guidelines to the “Introduction” section (see page 1): According to the Diagnostic and Statistical Manual of Mental Disorders (American Psychiatric Association, 2013), PE was delineated as a persistent or recurring pattern characterized by the occurrence of ejaculation within approximately one minute following vaginal penetration during partnered sexual activity, occurring prior to the volitional desire of the individual.
- In the "Introduction" section, in "1.1. PE and ED", the authors should also present data on the contemporary presence of both ED and PE, such as in PMID: 35915329.
Response: Thank you for your comment. We have added the following text and citation to the part of “Introduction” (see page 2): A recent study found that one in four men with ED have PE symptoms (Cilio et al., 2023).
- among penile girth augmentation techniques, I suggest also to report the benefits of glans penis augmentation, such as in PMID: 34400809.
Response: Thank you for your comment. We have added the following text and citation to the part of “Introduction” (see page 2): In addition, a recent review indicated that glans penis augmentation reduced PE symptoms (Califano et al., 2022).
- why did the authors choose 31 years to define young vs. old men? Usually, the scientific community uses 40 y.o. as a threshold.
Response: Thank you for your comment. There are two elements behind using 31 years as a cut-off between younger and older subjects. First, testosterone plays a role in every step of the male sexual response with previous research showing that lower testosterone levels are associated with reduced sexual desire and more ED. In addition, testosterone replacement treatment can improve the latency times of men with acquired PE (PE appearing only after a man’s first sexual experience). Both cross-sectional and longitudinal studies have shown that testosterone levels decline gradually as men age from their 30s to their 90s. The above text is now in the session of “Age and Sexual Dysfunction” (see page 3). Second, the age of 31 also happened to be the median in this sample. Based on the above, we decided to use 31 years as the cut-off. This element is now mentioned in the session “Statistical Analyses” (see page 7).
Reviewer 3 Report
Comments and Suggestions for Authors
I would like to thank the Editor for the opportunity to review this article.
The article "Associations Between Self-Reported Anatomical Characteristics of the Penis and Sexual Dysfunction in Men" is a well-written and informative manuscript.
The study sample size was large, which is a strength of the study. The manuscript is also well-written. The authors provide a clear introduction, methods section, results section, and discussion section.
I have a few specific comments that would improve the manuscript.
1. The introduction section is well-written and provides a good overview of the topic. However, it is too long and some of the information could be moved to the discussion section.
2. The method section is a bit limited. Can you provide more details about the recruitment process for the study?
3. Addresses of websites should not be in the text. It is distracting - for example in lines 249-250 websites should be added to references.
4. Tables 2, 3, and 4 present important data, but they could be better described. The authors should provide more information about the abbreviations and symbols used in the tables like B, SE, t.
5. In the questionnaire, the authors could ask participants about chronic diseases or medications they are taking. This information would be important to consider when interpreting the results of the study, as chronic diseases and medications can affect sexual function. What is more, in the discussion section, the authors could discuss the potential impact of chronic diseases and medications on sexual functioning.
I believe that these changes would improve the article and make it more informative.
Author Response
I would like to thank the Editor for the opportunity to review this article. The article "Associations Between Self-Reported Anatomical Characteristics of the Penis and Sexual Dysfunction in Men" is a well-written and informative manuscript. The study sample size was large, which is a strength of the study. The manuscript is also well-written. The authors provide a clear introduction, methods section, results section, and discussion section. I have a few specific comments that would improve the manuscript.
- The introduction section is well-written and provides a good overview of the topic. However, it is too long and some of the information could be moved to the discussion section.
Response: Thank you for your comment. We have moved some information in the “Introduction” section regarding penis size to the discussion section.
- The method section is a bit limited. Can you provide more details about the recruitment process for the study?
Response: Thank you for your comment. We have added more information about recruitment for the study. The text is the following: On both platforms, the platforms would send an invitation link of the questionnaire to potential participants who were registered on the platforms, that is, adult men who were 18-50 years old and currently living in Shanghai (see page 6).
- Addresses of websites should not be in the text. It is distracting - for example in lines 249-250 websites should be added to references.
Response: Thank you for your comment. We have deleted the addresses of the websites.
- Tables 2, 3, and 4 present important data, but they could be better described. The authors should provide more information about the abbreviations and symbols used in the tables like B, SE, t.
Response: Thank you for your comment. We have provided more information about the abbreviations and symbols including CHEES, IIEF-5, B, SE, t, and p. The test is the following: CHEES, Checklist for Early Ejaculation Symptoms; IIEF-5, International Index of Erectile Function–5; B, coefficient; SE, Standard Error; t, t-Test; p, probability (see pages 8, 9).
- In the questionnaire, the authors could ask participants about chronic diseases or medications they are taking. This information would be important to consider when interpreting the results of the study, as chronic diseases and medications can affect sexual function. What is more, in the discussion section, the authors could discuss the potential impact of chronic diseases and medications on sexual functioning. I believe that these changes would improve the article and make it more informative.
Response: Thank you for your comment. We added a regression analysis to investigate the association between SSRI medication status, cardiovascular and endocrine disease, CHEES (early ejaculation problems) and IIEF-5 (erection problems) (see page 11). The results indicated that currently using SSRI medication and currently having or having any history of cardiovascular and endocrine disease was associated with more early ejaculation and erectile problems. We have added the related result to the end of the “Results” section. We also added the related discussion for the result to the end of the “Discussion” section (see page 13). The text is the following: Interestingly, SSRI medication use was associated with both more PE and ED symptoms. The negative association between SSRI medication use and ejaculation control was not in line with previous studies which have found SSRI medication (i.e., fluoxetine, paroxetine) to improve the intervaginal ejaculation latency time in men with PE (Gul et al., 2022; Siroosbakht et al., 2019). In this sample, most participants using SSRI medication (27 in 29; 93%) were men without PE. Therefore, it may be that the underlying health problem (e.g. depression or anxiety) for which these men were taking SSRI may be behind the observed negative association. In contrast, it was not surprising to find a relationship between SSRI medication use and poorer erection given that one of the side effects of SSRI is the loss of sexual desire (Montejo et al., 2016; Siroosbakht et al., 2019).
Cardiovascular or endocrine diseases were found to be associated with more PE and ED symptoms. Previous studies have indicated that cardiovascular diseases were risk factors for ED (Pozzi et al., 2020), and also for premature ejaculation (Bolat et al., 2017). A previous study found that compared with males without sexual dysfunction, men with PE have higher levels of testosterone, and men with ED have higher levels of estradiol (Wu et al., 2016). The ratio of estradiol to testosterone was also higher in men with PE and ED compared with men without sexual dysfunction (Wu et al., 2016). Therefore, the underlying mechanism of the association between endocrine diseases and PE and ED symptoms may be the imbalance between different hormones.
Round 2
Reviewer 2 Report
Comments and Suggestions for Authors
Thank you for reviewing properly your article according to my suggestions.
Reviewer 3 Report
Comments and Suggestions for Authors
The authors have satisfactorily addressed most of my concerns.